# Near-Infrared Autofluorescence or Intraoperative Parathyroid Hormone Determination as a Surgical Support Tool in Primary Hyperparathyroidism: Too Close to Call?

**DOI:** 10.3390/cancers16234018

**Published:** 2024-11-30

**Authors:** Pietro Indelicato, Diego Barbieri, Emilio Salerno, Alberto Tettamanti, Mathilda Tersilla Guizzardi, Andrea Galli, Francesco Frattini, Leone Giordano, Mario Bussi, Gianlorenzo Dionigi

**Affiliations:** 1Otorhinolaryngology Unit, Division of Head and Neck Department, IRCCS San Raffaele Scientific Institute, 20132 Milan, Italy; indelicato.pietro@hsr.it (P.I.); barbieri.diego@hsr.it (D.B.); salerno.emilio@hsr.it (E.S.); tettamanti.alberto@hsr.it (A.T.); guizzardi.mathilda@hsr.it (M.T.G.); giordano.leone@hsr.it (L.G.); bussi.mario@hsr.it (M.B.); 2School of Medicine, Vita-Salute San Raffaele University, 20132 Milan, Italy; 3Division of Surgery, Istituto Auxologico Italiano IRCCS, 20145 Milan, Italy; f.frattini@auxologico.it (F.F.); g.dionigi@auxologico.it (G.D.); 4Department of Pathophysiology and Transplantation, University of Milan, 20122 Milan, Italy

**Keywords:** near-infrared autofluorescence (NIRAF), autofluorescence, primary hyperparathyroidism, parathyroid surgery, minimally invasive parathyroidectomy (MIP)

## Abstract

This study evaluates the effectiveness of near-infrared fluorescence imaging (NIFI) as a replacement for intraoperative parathyroid hormone (ioPTH) measurement during minimally invasive parathyroidectomy (MIP) in patients with primary hyperparathyroidism due to parathyroid adenoma. Fifty patients participated, undergoing preoperative imaging to locate the adenoma. The results indicated that MIP was successful in all cases, with a mean time to excision of the adenoma of 44.7 min and a total surgical duration of 85.5 min. The study also identified three distinct fluorescence patterns of the adenomas. Notably, when preoperative imaging confirms adenoma location, NIFI may reduce surgical time without compromising outcomes, suggesting a potential shift in surgical practice for PHP treatment.

## 1. Introduction

Primary hyperparathyroidism (PHP) is a common endocrine disorder characterised by excessive production of parathyroid hormone (PTH), usually by a solitary benign adenoma (85–90%), less commonly by multiple adenomas or hyperplasia of all four glands (5–10%), and very rarely by parathyroid carcinoma (<1%) [1]. Minimally invasive parathyroidectomy (MIP) is the recommended surgical treatment for a definitive cure for PHP. MIP is defined as any focused surgical procedure that aims to remove a single enlarged parathyroid gland, previously identified by preoperative exams (e.g., US and MIBI), limiting the unnecessary dissection of multiple glands or a bilateral cervical exploration [2]. Despite advances in preoperative imaging in terms of quality and sensitivity [3], the identification of the diseased parathyroid gland (PG) and thus the success rate of parathyroid surgery (95–99%) are highly dependent on the skill and experience of the surgeon [4]. Localisation of the parathyroid adenoma (PA) is the cornerstone of parathyroid surgery but can be challenging due to its similarity to the surrounding fatty tissue or its variable and unexpected location, especially when preoperative imaging is inconclusive or inconsistent [1].

Over the years, several intraoperative tools have been proposed to identify normal or abnormal PA, such as gamma probe, intraoperative ultrasound, frozen section, and intraoperative parathyroid hormone (ioPTH) measurement [2]. The use of ioPTH is based in particular on its short half-life (3 to 5 min): removal of the abnormal gland leads to a rapid decrease in circulating PTH levels and confirms the appropriateness of MIP [5]. However, some authors have questioned the use of ioPTH, as a similarly high cure rate after parathyroidectomy can be achieved without ioPTH if preoperative imaging (usually ultrasound and Tc99m pertechnetate/MIBI scan) clearly shows the location of the PA [6,7].

In recent years, a novel method for intraoperative identification of PGs has become available that is based on the detection of autofluorescence using near-infrared fluorescence imaging (NIFI) [8]. NIFI is the term referring to the imaging technique based on the intrinsic property of parathyroid tissue to emit light at a wavelength of 820 nm when irradiated by another near-infrared light (785 nm): this characteristic is called “autofluorescence” [8,9]. Although there is growing evidence of the reliability of NIFI in the identification and preservation of PG in thyroid surgery [10,11,12] and oncological surgery [13], the role of this technology in parathyroid surgery is still under investigation. NIFI can help the surgeon to differentiate PGs from other surrounding structures, but few studies have attempted to investigate quantifiable differences in fluorescence signals between normal and adenomatous PGs [4,14,15,16]. The main aim of this study is to assess whether the use of NIFI can assist the surgeon in identifying abnormal PGs. In particular, the aim is to investigate whether NIFI can replace ioPTH in the presence of two matching preoperative localisation images of the position of the PA, allowing a reduction in operating times without affecting the success rate. We aim to determine whether NIFI can offer comparable diagnostic accuracy and reliability in guiding surgical decision-making, thus potentially reducing the need for ioPTH monitoring in these specific cases. Another aim is to evaluate the different fluorescence patterns produced by adenomatous PGs and whether these can be used to further confirm the appropriate removal of PAs.

## 2. Materials and Methods

This clinical retrospective study included 50 patients who underwent MIP and were referred to the Department of Otorhinolaryngology of the San Raffaele Hospital (Milan, Italy) and to the Department of Surgery of the Istituto Auxologico Italiano (Milan, Italy) between March 2021 and April 2024. Informed consent was obtained from each patient for treatment and use of anonymised clinical data for study purposes. We obtained approval from the Institutional Review Board (IRB) of San Raffaele Hospital and Istituto Auxologico Italiano for this clinical trial with the protocol number 181/INT/2021, approved in July 2021. The study was conducted according to the ethical standards of the 1964 Declaration of Helsinki, as amended in 2020.

More specifically, patients with PHP caused by solitary PA underwent MIP with two concordant preoperative imaging techniques for localisation (ultrasound and Tc99m pertechnetate/MIBI scan), using an image-guided fluorescence measurement device (Fluobeam LX^®^, Fluoptics, Grenoble, France) and ioPTH measurement. The patients enrolled in this study were older than 18 years of age and spoke Italian as their mother tongue or were fully fluent in Italian. The diagnosis of PHP was made when elevated serum calcium (>2.60 mmol/L) and/or PTH levels (>65 pg/mL) were present. Patients with ambiguous or discordant preoperative localisation pictures, parathyroid carcinoma, or secondary, tertiary, MEN-related or familial hyperparathyroidism were excluded from the study. The following preoperative laboratory tests were performed on each patient: serum calcium, ionised calcium, PTH, creatinine, uraemia, 25-OH vitamin D, TSH, fT3 and fT4. In addition, all patients underwent ultrasound of the neck and Tc99m pertechnetate/MIBI examination. Preoperative imaging was considered concordant if both showed the exact location of the PA. MIP was performed in all patients using both NIFI and ioPTH. Frozen section analysis was never performed. All procedures were performed by the same surgeon for each institution. NIFI was used to assist the surgeon in identifying the PA during dissection and to evaluate the fluorescence pattern after resection of the specimen (Figure 1).

IoPTH was assessed according to the Vienna criterion [17,18], which considers a reduction in PTH level of at least 50% between a baseline value obtained from a blood sample taken before the surgical incision and a blood sample taken 10 min after PA removal to be significant. Each surgery was considered complete once the pathological PG was removed and confirmed by NIFI, but we still waited for PTH results after surgery to further confirm that the adenoma had been removed in all cases. Therefore, two different operation times were considered: the operation time until excision of the adenoma and the total operation time, including waiting for the ioPTH value after excision. In addition, the time between the surgical excision of the PA and the availability of the post-excision ioPTH result was evaluated.

The following intraoperative data were analysed: (1) the operative time until excision of the adenoma and the total operative time; (2) the time between PA excision and the availability of the post-excision ioPTH value; (3) the pattern of autofluorescence of the PA; (4) the pre- and post-excision PTH values; (5) the percentage deviation of the PTH value between pre- and post-excision; (6) the success rate according to the Vienna criterion. In each patient, serum calcium, ionised calcium and PTH levels were determined on the first postoperative day (POD) and one month after surgery. We considered any patient who had a total calcium level < 2.6 mmol/L (normal values: 2.1 mmol/L–2.6 mmol/L) and a PTH level < 65 pg/mL (normal values 15–65 pg/mL) during hospitalisation to be cured. Finally, the data on the need for calcium supplementation at discharge and the final histopathological report were evaluated.

## 3. Results

A total of 50 patients (46 female and 4 male) met the inclusion criteria for the study. All patients were followed up between 6 and 40 months after surgery (average 17.9 months). The average age of the cohort was 62.2 years (range 34–85 years) and the average BMI was 24.9 kg/m^2^ (SD ± 3.5 kg/m^2^). In total, 31/50 patients (62%) were asymptomatic, while 16/50 patients (32%) had kidney stones, and 19/50 patients (38%) had osteoporosis. Analysis of preoperative laboratory tests revealed a mean serum calcium of 3.2 mmol/L (SD ± 2.0 mmol/L), a mean ionised calcium of 1.4 mmol/L (SD ± 0.1 mmol/L) and a mean PTH of 164.5 pg/mL (SD ± 80.9 pg/mL). In all cases, neck ultrasonography and Tc99m pertechnetate/MIBI scan were concordant with the location of PA; i.e., all patients underwent MIP.

In 43/50 (86%), the PA was first identified by the surgeon and then confirmed by NIFI, while in 7/50 (14%) cases, the location of the PA was determined by NIFI prior to dissection and its visual identification. The mean duration of the surgical procedure until adenoma excision was 44.7 min (SD ± 25.2 min), while the mean duration of the entire procedure (including waiting for the result of the last intraoperative PTH value) was 85.5 min (SD ± 37.1 min). The variables “duration of the surgical procedure until adenoma excision” and “duration of the entire procedure” are NOT normally distributed (*p* < 0.05 in the Shapiro–Wilk test); therefore, we analysed the difference between the two variables by using non-parametric tests (i.e., tests that assess the difference in terms of median/IQR and not mean/SD). In this regard, the median of the duration of the surgical procedure until adenoma excision was 35.0 min (IQR 38.8), while the median duration of the entire procedure was 74.5 min (IQR 40.5). This difference in medians is statistically significant using the Wilcoxon non-parametric test for paired samples, *p* < 0.001.

The mean time between adenoma excision and the availability of the postoperative ioPTH value was 37 min (SD ± 12.2 min), totalling 1861 min.

In 23/50 cases (46%), PAs showed a single, well-defined, brightly fluorescent region, termed a “cap” by Demarchi et al. [19] (Figure 2A). In 15/50 cases (30%), the fluorescence pattern was heterogeneous (Figure 2B), while the remaining 12/50 PAs (24%) showed a homogeneous, hyperfluorescent pattern of autofluorescence (Figure 2C). We performed an anatomic–pathological analysis of the three different fluorescence patterns of the excised parathyroid glands, which showed that the hyperfluorescent area of the “cap” corresponds to healthy parathyroid parenchyma, while the less fluorescent area represents adenomatous tissue (Figure 3). In contrast, the samples that showed a heterogeneous or homogeneous hyperfluorescence pattern did not show a clear demarcation between healthy and adenomatous tissue, but rather a total infiltration of the glandular architecture by an adenoma or areas of healthy parenchyma mixed with adenomatous parenchyma (Figure 4).

The mean PTH level before surgery was 169.9 pg/mL (SD ± 85.1 pg/mL) and the mean PTH level after surgery was 37.0 pg/mL (SD ± 24.5 pg/mL), with a mean percentage variation in PTH level of 77.5% (SD ± 11.1%). In all patients, the ioPTH level decreased appropriately according to the Vienna criterion. All patients were successfully treated and had normal calcium (mean 2.4 mmol/L, SD ± 0.2 mmol/L) and PTH (mean 26.2 pg/mL, SD ± 17.3 pg/mL) levels on the first day of treatment. Calcium supplementation was required at discharge in 43/50 (86) of patients (1.5 g calcium carbonate per day). The pathological findings confirmed the diagnosis of PA in all patients. The mean PA diameter was 13.2 mm (SD ± 4.8 mm) and the mean weight was 1.1 g (SD ± 1.2 g). One month after surgery, all patients had normal calcium and PTH levels. All surgical details are described in Table 1.

## 4. Discussion

The use of NIFI has become widespread to assist surgeons in the early detection of normal or abnormal PG. Although there is increasing evidence in thyroid surgery of the ability of NIFI to preserve the integrity of PGs and prevent transient hypocalcaemia [10], the role of autofluorescence in parathyroid surgery is still under investigation. The main aim of this study is to investigate whether the intraoperative use of autofluorescence in patients with a single, colocalised parathyroid adenoma can replace the use of ioPTH monitoring by at least two imaging examinations (usually neck ultrasound and scintigraphy with Tc99m-pertecnetate/MIBI) without compromising success rates. Our main result showed that approximately 1861 min (average 37 min per patient) of surgery time would have been saved if confirmation of correct PA removal had been performed with NIFI only and without ioPTH measurement. We nevertheless decided to use ioPTH to further confirm the correct surgical procedure, as we chose not to compare our patient group with a control group in which autofluorescence was not used. The reliability of autofluorescence is confirmed by an appropriate reduction in ioPTH in each patient.

There is no consensus on the use of intraoperative parathyroid hormone (ioPTH) measurement in parathyroid surgery. Some studies suggest it can be omitted in patients with concordant preoperative imaging, while others report that it improves success rates [5,20,21,22,23,24,25,26,27,28]. Mihai et al. found similar outcomes in patients with and without ioPTH monitoring during minimally invasive parathyroidectomy (MIP) [20]. Similarly, Sartori et al. observed no significant difference in cure rates between groups with and without ioPTH [5]. In contrast, Riss et al. reported a decreased cure rate and increased disease persistence in patients without ioPTH [23,24]. A systematic review by Quinn et al. found higher cure rates with ioPTH, but only marginal benefits when preoperative imaging was concordant [26]. Disadvantages of ioPTH include test cost, time delays, and potential false negatives leading to unnecessary exploration. Cost-effectiveness analyses suggest that ioPTH increases success rates modestly and is cost-effective if the test is under USD 110 or revision surgery costs exceed USD 12,000 [29].

Another aim of this study is to evaluate the role of autofluorescence in parathyroid surgery, focusing on the ability of NIFI to detect PAs and their major autofluorescence patterns early. Our results showed that in 86% of patients, the location of the parathyroid gland was first identified by the surgeon and then confirmed by autofluorescence, while the adenoma was detected by autofluorescence in only 14 cases and only then by the surgeon’s naked eye. Therefore, it can be assumed that autofluorescence, similar to ioPTH, plays a role in confirming the correct excision of PA and not in its early detection. Regarding the autofluorescence pattern, we found that most PAs (23/50 cases, 46%) showed the presence of a cap (Figure 2A), while the rest showed a heterogeneous (15/50 cases, 30%) (Figure 2B) or homogeneous hyperfluorescent pattern (12/50 cases, 24%) (Figure 2C).

As mentioned above, the fluorescence pattern of PAs is not well defined, and the results of different studies are inconsistent. Some authors claim that PAs are more fluorescent than normal PGs [30], whereas other authors claim that PAs are darker and more heterogeneous [31]. A recent study of 131 parathyroidectomies by Lee et al. showed that the mean intensity of near-infrared autofluorescence (NIR-AF) of PGs had a negative correlation with weight (the brighter the gland, the brighter the fluorescent signal; *p* = 0.019) and a positive correlation with age (the older the patient, the brighter the gland; *p* = 0.013). The authors found no significant correlations with preoperative serum calcium and PTH, body mass index or gender (*p* > 0.05) [14]. Another seminal study by Demarchi et al. showed that PAs had a cap region with a well-defined bright fluorescent area at the top of the gland in about 74% of cases, while the remaining 26% of patients had a weak or heterogeneous signal [19]. In addition, the authors performed a histological analysis of the hyperfluorescent area and found that the fluorescent “cap” corresponded to a rim of normal parathyroid tissue in 15 of 17 samples [14]. Since it is not known which endogenous fluorophore is responsible for the autofluorescence of the parathyroid gland, there is no clear correlation between the fluorescence pattern and the histological results of the sample. The first fluorophore candidate was the extracellular calcium receptor (CasR), which is present in the highest concentration in the parathyroid tissues and in lower concentrations in the thyroid gland and is not present in other neck tissues [32]. The presence of hyperplastic or adenomatous parathyroid cells, patchy fibrosis areas, oxyphilic cell clusters and haematoma areas could be responsible for the reduction in CasR expression in primary hyperparathyroidism and consequently for the lower brightness of the fluorescence signals [31,32,33].

The use of autofluorescence in parathyroid surgery has advantages and disadvantages. The main advantage is the short learning curve required to use this technology. The software interface and the device are user-friendly and allow for quick understanding and application after a few procedures. The use of NIFI is safe. There are no studies in the literature reporting complications with the use of this technology, as no dyes or contrast agents need to be used. Although autofluorescence detection devices have a significant initial cost, the only consumables required are the sterile sheaths that cover the camera, which are available at affordable prices. However, there are some limitations when using this technology. Firstly, the laser depth does not go beyond 2–3 mm, which means that the PGs are not visible outside the skin or when embedded in fatty or connective tissue. Secondly, colloidal thyroid nodules, pathological lymph nodes or patients with chronic thyroiditis may show higher fluorescence under near-infrared light than the background, leading to false positive results. Thirdly, the duration of the procedure was prolonged, especially during the first few operations, because the operating lights had to be directed away from the surgery field each time [11]. Finally, there are no data in the literature to date on the ability of autofluorescence to distinguish a single adenoma from multiglandular hyperplasia, which is the most challenging aspect of this procedure. The greatest strength of our manuscript is that it is one of the first studies to attempt to characterise the fluorescence of parathyroid adenomas qualitatively (cap, homogeneous, heterogeneous) rather than quantitatively (brighter or darker). Moreover, this is the first study in our country to report the use of autofluorescence in primary hyperparathyroidism. However, some limitations should be noted: the retrospective design and the small number of patients involved in this preliminary report do not allow us to draw conclusions about the reliability of autofluorescence in parathyroid surgery. We endeavoured to produce a preliminary study in non-complex cases (co-localised adenomas) to validate the accuracy of NIFI. Future prospective studies will be needed to test the efficacy of this technology even in challenging cases, for example, for those adenomas that have not been successfully localised with preoperative imaging. Another limitation is that the device available in our institution does not allow for a quantitative analysis; only qualitative data can be obtained. We are currently waiting for a new software update that will provide a quantitative evaluation of fluorescence. A prospective study will then be necessary to gather new data based on the updated technology.

## 5. Conclusions

Despite the advances that have been made in preoperative imaging for localisation, the identification of PA and therefore the success of parathyroid surgery still depend on the surgeon’s experience. If ultrasound and Tc99m pertechnetate/MIBI scan are concordant in localising the PA, autofluorescence devices can be considered as safe tools and used instead of ioPTH, allowing the operation time to be shortened without decreasing the success rate. In addition, the use of autofluorescence devices could play an even more important role in more complex cases of single PA where preoperative investigations are inconclusive or inconsistent. Finally, future research should shed light on the role of autofluorescence in differentiating between adenoma and hyperplasia. Further prospective or multicentre studies should be performed to validate these results and the accuracy of this technology in parathyroid surgery.

## Figures and Tables

**Figure 1 cancers-16-04018-f001:**
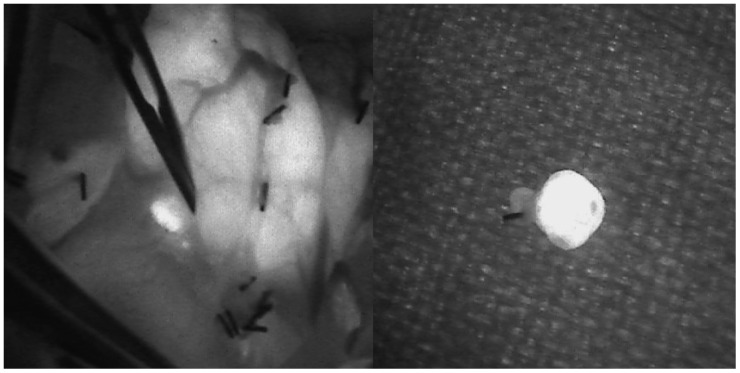
Intraoperative identification of a right inferior parathyroid adenoma under near-infrared light (**left**); ex vivo inspection of a resected parathyroid adenoma with near-infrared autofluorescence (**right**).

**Figure 2 cancers-16-04018-f002:**
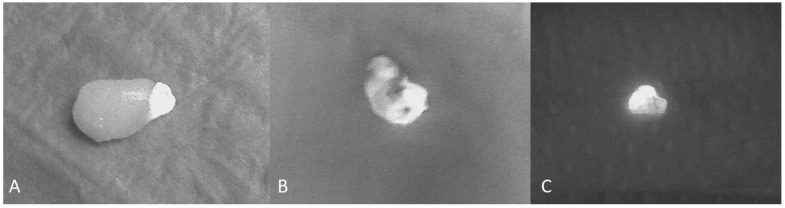
Three different patterns of fluorescence of excised parathyroid adenomas: single well-defined fluorescent area on the top of the specimen (**A**); heterogeneous (**B**) and homogeneous hyperfluorescence (**C**).

**Figure 3 cancers-16-04018-f003:**
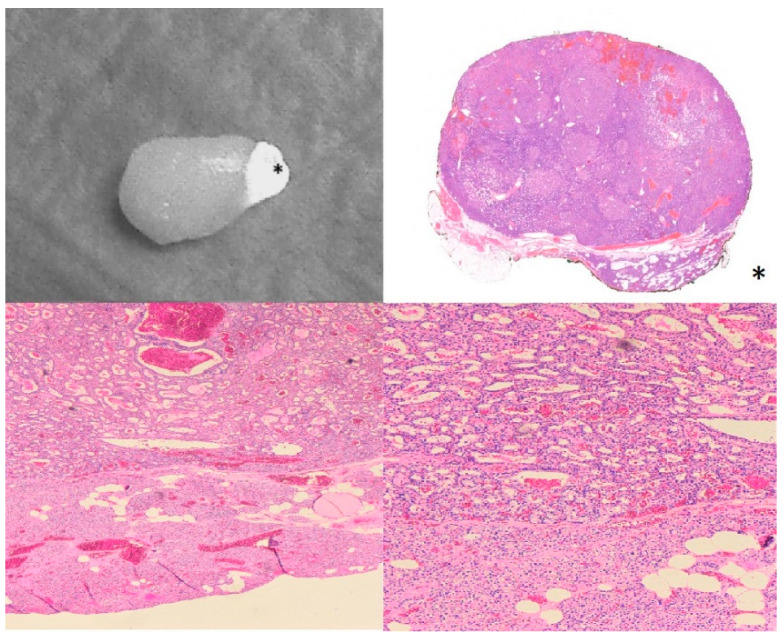
Comparison between near-infrared fluorescence imaging (**upper-left**) and histological imaging (**upper-right**) of the same parathyroid adenoma. The black asterisk (*) highlights the cap of fluorescence which corresponds to the non-adenomatous parenchyma of the resected gland. The area without a significant fluorescent signal corresponds to the adenoma. At the bottom, two histological images at two different magnifications of the interface between adenoma and normal parathyroid gland parenchyma are shown (magnification, 4× and 10×).

**Figure 4 cancers-16-04018-f004:**
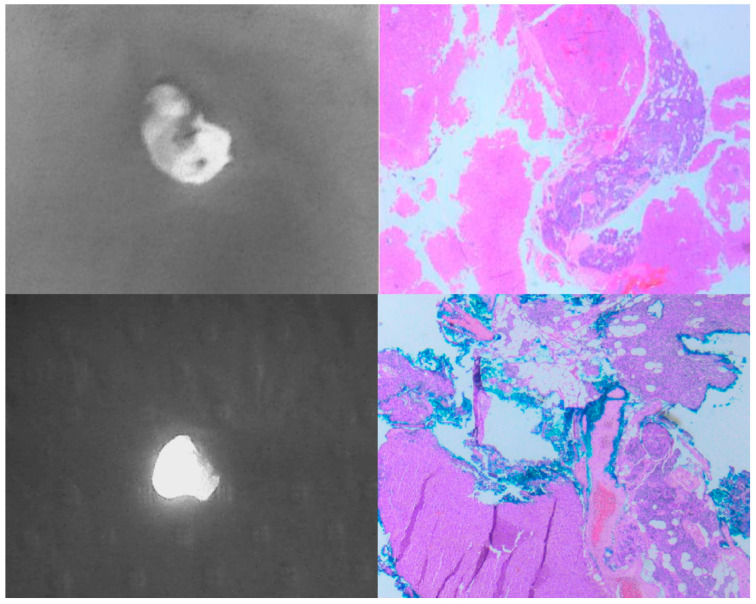
Two parathyroid adenomas showing heterogeneous (**upper-left**) and homogeneous hyperfluorescent (**bottom-left**) pattern of fluorescence. The corresponding histological images (**upper-right** and **bottom right**) showing areas of healthy parathyroid tissue mixed with adenomatous parenchyma (magnification, 10×).

**Table 1 cancers-16-04018-t001:** Intraoperative and postoperative results. SD: standard deviation; NIFI: near-infrared fluorescent imaging; ioPTH: intraoperative PTH; POD: postoperative day; PA: parathyroid adenoma.

Parameters	Mean (±SD)
PA identified with NIFI before visual inspection n°/tot (%)	7/50 (14%)
Surgical time at the excision of the adenoma (minutes)	44.7 (±25.2)
Overall surgical time (minutes)	85.5 (±37.1)
Mean time between PA excision and ioPTH availability (minutes)	37 (±12.2)
Pattern of fluorescence of PAs	
Cap n°/tot (%) Heterogeneous pattern n°/tot (%) Homogeneous pattern n°/tot (%)	23/50 (46%) 15/50 (30%) 12/50 (24%)
Pre-excision PTH level (pg/mL)	169.9 (±85.1)
Post-excision PTH level (pg/mL)	37.0 (±24.5)
Variation in PTH level (%)	77.5 (±11.1)
Serum calcium level POD-1 (mmol/L)	2.4 (±0.2)
Serum PTH level POD-1 (pg/mL)	26.2 (±17.3)
Patients who need calcium supplementation at discharge n°/tot (%)	43/50 (86%)
PA diameter (mm)	13.2 (±4.8)

## Data Availability

Data are contained within the article.

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
