# Peer review of "Near-Infrared Autofluorescence or Intraoperative Parathyroid Hormone Determination as a Surgical Support Tool in Primary Hyperparathyroidism: Too Close to Call?"

_cancers, 2024, doi:10.3390/cancers16234018_

Round 1
Reviewer 1 Report
Comments and Suggestions for Authors
I red with interest the article, “Near-infrared autofluorescence or intraoperative parathyroid hormone determination as surgical support tools in primary hyperparathyroidism: too close to call?” is an insightful study that evaluates the potential of near-infrared fluorescence imaging (NIFI) as an alternative to intraoperative parathyroid hormone (ioPTH) measurement in primary hyperparathyroidism (PHP) surgery. Involving 50 patients, the study assesses the efficacy and time efficiency of NIFI for adenoma localization/removal confirmation during minimally invasive parathyroidectomy (MIP), offering valuable insights into fluorescence patterns and their potential to reduce operating times when preoperative imaging supports adenoma localization.
Here some points that should be improved:
• Clarification on MIP: Please provide a clear definition of minimally invasive parathyroidectomy (MIP) early in the article.
• Consistency in Terms: Terms such as “NIFI” and “autofluorescence” are used interchangeably, which may be confusing for readers unfamiliar with the technology. For consistency, choose one term (e.g., NIFI) and define it clearly at the outset. This will help readers follow the methods and results more easily.
• Line 101: You state that “All procedures were performed by the same surgeon.” How was this achieved across two unrelated facilities? Please clarify.
• Line 107-108: The term “expected” is unclear. Could you specify what is expected here?
• Figure 3: Please place the * symbol on the “cap” as mentioned in the legend for better clarity.
• Ethical Approval: Was this study approved by an ethics committee? If applicable, please add the ethical approval number to the paper.
Patient Selection and Bias
The most significant potential bias in this study arises from patient selection. Since the patients were pre-selected for single adenomas based on preoperative imaging, ioPTH may not be strictly necessary in this specific subset. However, I agree with the value of confirming adenoma localization with NIFI, which is not time-consuming.
The presence of a “cap” may suggest a monoclonal growth of parathyroid cells forming an adenoma, whereas a heterogeneous NIRAF signal could indicate polyclonal cell growth, potentially compatible with multiglandular disease Do you found differences in histological diagnosis?
The study’s hypothesis would benefit from greater clarity, especially regarding whether the goal is to assess NIFI as solely an adjunctive tool or as a possible replacement for ioPTH in cases with conclusive preoperative imaging. A more concise restatement of this objective in the introduction could improve clarity.
The study sample (50 patients) is relatively small, which may limit the generalizability of the fluorescence patterns and outcomes. Acknowledging this limitation
The article introduces different fluorescence pattern types (cap, heterogeneous, homogeneous) but does not provide a quantitative analysis or rationale for these categories please incorporate quantitative measures
Although the results suggest that NIFI may reduce surgical time the article does not clarify whether these time savings are statistically significant or anecdotal. Include statistical analysis to validate the claimed reduction in operating time with NIFI.
The article mentions cost considerations of ioPTH testing but does not provide a comparative cost analysis for implementing NIFI. Include the cost implications of adopting NIFI as a routine practice and its long-term feasibility
Minor Corrections:
There are minor typographical errors, such as inconsistent use of “ioPTH” versus “iPTH,” and some grammatical inaccuracies. Ensuring consistency and correcting these errors will improve readability.
The study provides a valuable exploration of NIFI as a potentially beneficial tool in parathyroid surgery particularly for reducing dependence on ioPTH in cases of confirmed adenoma localization. With adjustments for clarity, statistical support for time efficiency, and further discussion on cost and sample size limitations, this article could make a contribution to discussions on the evolving role of intraoperative imaging techniques in endocrine surgery.
Author Response
Clarification on MIP: Please provide a clear definition of minimally invasive parathyroidectomy (MIP) early in the article.
Answer: Thank you for your suggestion. We have modified the manuscript adding the definition of MIP (minimally invasive parathyroidectomy (MIP) is defined as any focused surgical procedure that aims to remove a single enlarged parathyroid gland, previously identified by preoperative exams (e.g. US and MIBI), limiting the unnecessary dissection of multiple glands or a bilateral cervical exploration). Lines 47-50.
Consistency in Terms: Terms such as “NIFI” and “autofluorescence” are used interchangeably, which may be confusing for readers unfamiliar with the technology. For consistency, choose one term (e.g., NIFI) and define it clearly at the outset. This will help readers follow the methods and results more easily.
Answer: Thank you for the remark. We have clarified the difference between the terms NIFI and autoflorescence; in particular, NIFI (near infrared fluorescent imaging) is the term referring to the imaging technique based on the intrinsic property of parathyroid tissue to emit light at a wavelength of 820 nm when irradiated by another near-infrared light (785 nm) which is is called “autoflorescence”. Lines 68-71.
Line 101: You state that “All procedures were performed by the same surgeon.” How was this achieved across two unrelated facilities? Please clarify.
Answer: Thank you for the suggestion. We rephrased the sentence and corrected it: all the procedures were performed by the same surgeon for each institution. Line 110-111.
Line 107-108: The term “expected” is unclear. Could you specify what is expected here?
Answer: Thank you for the observation, we have changed the phrase to make clearer what we “expected” (which meant “what we thought would happen, based on our previous knowledge” in the context of the article). Lines 118-119.
Figure 3: Place the * symbol on the “cap” as mentioned in the legend for better clarity.
Answer: Thank you. We have modified the manuscript accordingly. (Figure 3)
Ethical Approval: Was this study approved by an ethics committee? If applicable, please add the ethical approval number to the paper.
Answer: Thank you for the observation. The protocol number for the ethical committee approval was actually included in the section “Institutional Review Board Statement”, we have also included it in the materials and methods section for completion (lines 92-95).
Patient Selection and Bias
The most significant potential bias in this study arises from patient selection. Since the patients were pre-selected for single adenomas based on preoperative imaging, ioPTH may not be strictly necessary in this specific subset. However, I agree with the value of confirming adenoma localization with NIFI, which is not time-consuming.
Thank you for the comment. As a review of existing literature reveals, the use of NIFI has been considered to be beneficial even in instances where the adenoma has been co-localised. The efficacy of NIFI must still be subjected to rigorous testing in order to prove its worth in challenging cases and the current study is a preliminary investigation which aims to validate this technology. We agree with your observation and we believe that future prospective studies may shine some light on the matter.
We have proceeded to include a summary of this paragraph in the manuscript in our discussion section concerning possible limitations (lines 295-299).
The presence of a “cap” may suggest a monoclonal growth of parathyroid cells forming an adenoma, whereas a heterogeneous NIRAF signal could indicate polyclonal cell growth, potentially compatible with multiglandular disease. Did you find differences in histological diagnosis?
Answer: Thank you for your thoughtful question. Upon reviewing the histological diagnoses in relation to the NIRAF signal characteristics, we did not observe significant differences that would distinguish between monoclonal and polyclonal growth patterns. While the presence of a "cap" in imaging might suggest a monoclonal growth, typically seen in parathyroid adenomas, and a heterogeneous NIRAF signal might imply polyclonal growth indicative of multiglandular disease, our histological analysis did not reveal a clear distinction between these patterns.
We have included a few lines discussing this matter in the manuscript (lines 259-264).
The study’s hypothesis would benefit from greater clarity, especially regarding whether the goal is to assess NIFI as solely an adjunctive tool or as a possible replacement for ioPTH in cases with conclusive preoperative imaging. A more concise restatement of this objective in the introduction could improve clarity.
Answer: Thank you for your constructive feedback. We appreciate your suggestion to clarify the study's hypothesis, and we agree that a more explicit statement would enhance the overall clarity of the manuscript.
The primary objective of this study is to evaluate the potential of NIFI as a replacement for intraoperative parathyroid hormone (ioPTH) measurement, particularly in cases where preoperative imaging provides conclusive localization of the parathyroid adenoma. We aim to determine whether NIFI can offer comparable diagnostic accuracy and reliability in guiding surgical decision-making, thus potentially reducing the need for ioPTH monitoring in these specific cases.
To address your concern, we have revised the introduction to more clearly articulate this hypothesis and more precisely define the role of NIFI in the context of preoperative imaging. We believe this will improve the clarity of our objectives and strengthen the focus of the study. You can find our changes in lines 78-83 of the manuscript.
The study sample (50 patients) is relatively small, which may limit the generalizability of the fluorescence patterns and outcomes. Acknowledging this limitation.
Answer: Thank you for the observation, we are aware of the limitation concerning the small sample size and the suggestion is greatly appreciated. We included the remark in our discussion (lines 293-295).
The article introduces different fluorescence pattern types (cap, heterogeneous, homogeneous) but does not provide a quantitative analysis or rationale for these categories please incorporate quantitative measures.
Answer:
Dear Reviewer, the fluobeam device we possess does not allow for a quantitative analysis, only qualitative data can be obtained. We are currently waiting for a new software update that will provide for a quantitative evaluation of fluorescence. A prospective study will then be necessary to gather new data based on the updated technology.
I hope this response addresses your question, we included it in our limitations (lines 299-304). Please feel free to reach out if you need further clarification.
Although the results suggest that NIFI may reduce surgical time the article does not clarify whether these time savings are statistically significant or anecdotal. Include statistical analysis to validate the claimed reduction in operating time with NIFI.
Answer: Thank you for the excellent observation. We have provided and included a statistical analysis in order to address your concern. The results are the following:
The variables “Surgical time to excision” and “Total surgical time including waiting” are NOT normally distributed (p < 0.05 in the Shapiro-Wilk test), therefore, the difference between the two variables should be analyzed using NON-parametric tests (i.e., tests that assess the difference in terms of median-IQR and not mean-SD).
In this regard:
- Surgical time to excision: median 35.0 min (IQR 38.8)
- Total surgical time including waiting: median 74.5 min (IQR 40.5)
This difference in medians is statistically significant using the Wilcoxon non-parametric test for paired samples, p < 0.001.
We have included the analysis in our “results” section (lines 151-159).
The article mentions cost considerations of ioPTH testing but does not provide a comparative cost analysis for implementing NIFI. Include the cost implications of adopting NIFI as a routine practice and its long-term feasibility
Answer: Dear Reviewer, thank you for your observation. The cost of the Fluobeam is contingent upon the initial purchase of the machine, which can be utilized in a multitude of surgical procedures, including thyroid and reconstructive oncological surgeries. This allows for the initial cost to be amortized over time. Once the machine is purchased, the cost per procedure is dependent upon the use of sterile covers.
This remark is summarised at lines 276-278 of our discussion section.
Minor Corrections:
There are minor typographical errors, such as inconsistent use of “ioPTH” versus “iPTH,” and some grammatical inaccuracies. Ensuring consistency and correcting these errors will improve readability.
Answer: Thank you kindly for the observation. We have modified the manuscript accordingly, using only ioPTH and ensured consistency.
Reviewer 2 Report
Comments and Suggestions for Authors
The manuscript of Indelicato et al. investigated the use of near-infrared fluorescence imaging (NIFI) compared with intraoperative PTH measurement in a series of 50 patients undergoing parathyroidectomy for the diagnosis of primary hyperparathyroidism. The item and the results are of interest.
Major concerns:
Methods:
- Page 3 Line 129: Which criteria have been used to define asymptomatic vs symptomatic PHPT ?
- Authors should provide evaluation of the statistical power for the main endpoint (NIFI can replace the use of ioPTH monitoring?) given the size of the investigated sample (n=50)
- Number and date of the protocol approval by ethical committee should be provided
Results:
- Is there any significant difference in histology of PA s with the different pattern of NIFI positivity? Chief cells versus oxyphilic cells?
Discussion:
- the paragraph (lines 194-225) describing results of the studies investigating ioPTH needs to be shorten
- The anatomic-pathological analysis of the three different fluorescence patterns of the excised parathyroid glands (lines 249-256) should be moved in the section Results as it is not appropriate in the section Discussion
- The Authors should highlight that the results of the present study are limited to PHPT patients with a single, colocalized parathyroid adenoma; the inclusion criteria adopted by the Authors excluded “challenging” patients where parathyroid tumors are not localized or their localization is discordant among the different imaging procedures.
Author Response
Methods
Page 3 Line 129: Which criteria have been used to define asymptomatic vs symptomatic PHPT?
Answer: Dear author, to define asymptomatic or symptomatic PHPT we use the definition provided by the Fifth International Workshop on Primary hyperparathyroidism (Bilezikian JP, Khan AA, Silverberg SJ, Fuleihan GE, Marcocci C, Minisola S, Perrier N, Sitges-Serra A, Thakker RV, Guyatt G, Mannstadt M, Potts JT, Clarke BL, Brandi ML; International Workshop on Primary Hyperparathyroidism. Evaluation and Management of Primary Hyperparathyroidism: Summary Statement and Guidelines from the Fifth International Workshop. J Bone Miner Res. 2022 Nov;37(11):2293-2314. doi: 10.1002/jbmr.4677. Epub 2022 Oct 17. PMID: 36245251).
Authors should provide evaluation of the statistical power for the main endpoint (NIFI can replace the use of ioPTH monitoring?) given the size of the investigated sample (n=50)
Answer: Thank you for the observation. We have obtained a statistical analysis which we have included in the paper in our results section (lines 151-159). We hope the addition provides clarification for your question.
Number and date of the protocol approval by ethical committee should be provided
Answer: Dear Reviewer, we have included the information you have requested in our “materials and methods” subsection specifically lines 91-95.
Results
Is there any significant difference in histology of PA s with the different pattern of NIFI positivity? Chief cells versus oxyphilic cells?
Answer: Thank you for your excellent question. Upon reviewing the histological diagnoses in relation to the NIRAF signal characteristics, we did not observe significant differences that would distinguish between monoclonal and polyclonal growth patterns. While the presence of a "cap" in imaging might suggest a monoclonal growth, typically seen in parathyroid adenomas, and a heterogeneous NIRAF signal might imply polyclonal growth indicative of multiglandular disease, our histological analysis did not reveal a clear distinction between these patterns. I hope this response provides a satisfactory answer to your question. We provided a few lines on the subject in the revised manuscript (lines 259-264).
The paragraph (lines 194-225) describing results of the studies investigating ioPTH needs to be shorten
Answer: Thank you for the observation. We have endeavored to modify the manuscript accordingly by shortening the paragraph and making it more clear (lines 224-236).
The anatomic-pathological analysis of the three different fluorescence patterns of the excised parathyroid glands (lines 249-256) should be moved in the section Results as it is not appropriate in the section Discussion
Answer: Thank you for the excellent point. We have modified the manuscript accordingly and moved the paragraph in question to lines 165-173 of the revised manuscript.
The Authors should highlight that the results of the present study are limited to PHPT patients with a single, colocalized parathyroid adenoma; the inclusion criteria adopted by the Authors excluded “challenging” patients where parathyroid tumors are not localized or their localization is discordant among the different imaging procedures.
Answer: Thank you for the observation. We included in our discussion our reasoning for choosing co-localised cases of parathyroid adenoma at lines 296-299. We hope this provides a clarification for the question you have raised.